

# A flood risk oriented dynamic protection motivation framework to explain risk reduction behaviours

Philippe Weyrich[1], Elena Mondino[2,3], Marco Borga[4], Giuliano Di Baldassarre[2,3], Anthony Patt[1], Anna Scolobig[5]

[1] Climate Policy Group, Department of Environmental Systems Science, Swiss Federal Institute of Technology (ETH Zurich), 8092 Zurich, Switzerland
[2] Department for Earth Sciences, Uppsala University, 75236 Uppsala, Sweden
[3] Centre of Natural Hazards and Disaster Science, Sweden
[4] Department of Land, Environment, Agriculture and Forestry, University of Padova, 35122 Padova, Italy
[5] Environmental Governance and Territorial Development Institute, University of Geneva, 1205 Geneva, Switzerland

*Correspondence to*: Philippe Weyrich (philippe.weyrich@usys.ethz.ch)

**Abstract.** Private risk reduction behaviours can significantly reduce the negative impacts of flooding and flash floods. Over the past decades, researchers have used various socio-cognitive models or threat/coping mechanisms to explain individual protective behaviours. However, these models ignore that people are not equally ready to act upon a danger and they give limited insights into the effectiveness of communication strategies to foster risk reduction behaviours. Therefore, we explored the current state of homeowner's readiness to undertake risk reduction behaviours in flood risk areas by applying a dynamic protection motivation framework. We conducted a survey in an Italian municipality that experienced severe flash flooding in September 2018. The results show that people are motivated by different factors in prompting risk reduction behaviour based on their type of protective measures. For example, people that undertook structural or avoidance measures are more likely to be motivated to protect themselves by increased perceptions of vulnerability and response efficacy, and are less worried about expected flood losses compared to people that undertook only basic emergency measures. In this paper, we argue how these new insights contribute to target flood risk communication strategies to groups of individuals characterized by different readiness stages and motivations to protect themselves.



# 1 Introduction

Flooding and flash floods are one of the major causes of natural hazards related deaths, they affect the life and safety of millions of people worldwide (Guha-Sapir et al., 2017; Kellens et al., 2013). In 2017 alone, 345 flood events killed 6500 people and caused economic losses of US$ 42.5 billion (Munich RE, 2017). With climate change influencing extreme weather events, more people will likely be exposed to higher flood risk in the near future than they are at present (IPCC, 2012, 2014). For example, most cities worldwide are projected to experience an increase of precipitation rates in the form of heavy rainfall (IPCC, 2012). The high proportion of sealed surfaces further increases flood risk (Carter, 2011), and the concentration of people, assets, critical infrastructure, as well as political and economic activities exacerbates the impacts of extreme events (European Environment Agency, 2017). Hence, the protection of livelihoods from flood events as an adaptation strategy to climate change will be of high priority in the coming years. Different types of adaptation to flood risk exist. For social systems one can differentiate between administrative and private adaptation, as well as between reactive adaptation during a flood and precautionary adaptation before a flood (Grothmann and Reusswig, 2006). Given similar exposure and sensitivity to floods (that, along with adaptation, determine flood damage and vulnerability), private protective action can have significant effects on flood outcomes as it was documented, for instance in Grothmann and Reusswig (2006). Risk reduction behaviours include structural measures (e.g. special installation of heating and electric system, anti-backflow valves), avoidance measures (e.g. keeping valuables and expensive appliances above flood-prone areas of the house) and emergency preparedness (e.g. own sandbags available, emergency plan at hand).

The adoption of risk reduction measures plays a critical role in the transition from top-down to people-centred approaches for disaster risk management (Scolobig et al., 2015). Authorities and public officials highlight the importance of people protecting themselves and see sharing responsibility as an appropriate form of response (Bostrom et al., 2015; Box et al., 2013; Morss et al., 2015; Scolobig et al., 2015). However, it remains unclear whether this actually matches the capacity of individuals to increase their level of protection (Scolobig et al., 2015). For instance, in the case of floods, citizens often transfer the responsibility for their own safety and protection to the agencies in charge (Adger et al., 2013; Bichard and Kazmierczak, 2012). This clearly reveals that one of the key challenges in the transition of people-centred approaches is the responsibility shift from the authorities to the public, demanding that the latter take precautionary actions (Scolobig et al., 2015). In face of this change, it is important to investigate the current readiness of individuals to undertake risk reduction behaviours and to identify the triggers that influence individuals to act.

So far, researchers have used socio-cognitive models to explain risk reduction behaviours (Grothmann and Patt, 2005; Grothmann and Reusswig, 2006). The most important factors used in these models are risk perception, hazard experience, risk aversion and socio-demographics. The influence of coping mechanisms was highlighted in studies that followed the Protection Motivation Theory (PMT) (Kellens et al., 2013; Poussin et al., 2014). More recently, the role of insurance and public assistance on private risk reduction behaviour have been investigated (Hanger et al., 2018). All these models have in common that they



assume that people are equally ready to act upon danger and they give limited insights into the effectiveness of communication strategies to foster risk-reduction behaviour. However, the public is no homogeneous group (Martens et al., 2009). Some people might be ready to take responsibility and act to reduce their flood risk exposure, while others adopt different behaviours. Thus, there are different stages or degrees of readiness which can influence individual motivation and intention to protect

themselves from a risk (Horwath, 1999; Prochaska et al., 1995). In the Trans-Theoretical Model (TTM), people are assigned to one of several stages based on their behaviours and intentions to undertake risk reduction actions (Prochaska et al., 1995). For example, Horwath (1999) addressed the question whether eating behavioural change follows a stage process, that could be used to help nutritionists to identify the predominant stages in a population. Enhanced knowledge about these stages allows to focus resources on the issues that will most likely move people to the next stage.  In medical research, Block and Keller

(1998) showed that people are at different stages of readiness to undergo medical testing, which can be affected by cognitive processes.

In this study, we focus on the drivers of risk reduction behaviours, employing a dynamic framework that combines the Protection Motivation Theory (PMT) and the Trans-Theoretical Model (TTM). The aim of this study is two-fold. First, we aim to determine the usefulness of this dynamic protection motivation framework to understand people's perception of flood risk

and their readiness to undertake risk reduction behaviours. Second, we aim to identify the most important factors in prompting behaviours in different risk reduction stages. This categorization allows us to identify the key issues to improve communication with people in each stage. Thus, we contribute to target flood risk communication strategies to groups of individuals characterized by different readiness stages and motivations to protect themselves.

## 2 Background

Flood management has shifted from a primarily top-down, command and control, approach to an integrated approach that increasingly addresses the role of private households in implementing flood damage mitigation measures (Kellens et al., 2013; Terpstra and Gutteling, 2008). To foster this transition, there is a need to integrate the analysis of resident´s risk perception and preparedness into local flood risk management practices (Figueiredo et al., 2009). The majority of studies have focussed on flood risk perceptions, for instance via the effect of past experience and affect (Keller et al., 2006; Siegrist and Gutscher,

2006, 2008), or as a mean to explain and promote private protective behaviours (Kellens et al., 2013). However, this is not always supported on theoretical or empirical grounds as Kellens et al. (2013) document: higher risk perceptions alone do not necessarily lead to a positive change in protective behaviour. For example, Scolobig et al. (2012) found no link between resident´s level of risk awareness and preparedness. Wachinger et al. (2013) found that high risk perceptions (predominantly shaped by personal experience and trust) do not lead to risk reduction behaviours if people accept the risk (i.e. benefits that

outweigh costs), if they do not feel responsible or if they have little resources. Other recent studies highlight instead that protective behaviours are not only influenced by high risk perceptions, but also by high coping appraisal (a person's perceived



ability to cope with and avoid being harmed by a threat in an effective way) (Bubeck et al., 2012; Kellens et al., 2013; Parker et al., 2009). These results, in part contradictory, show that it is important to not only focus on flood risk perception as the main driver of risk reduction behaviour, but also to understand other factors that might trigger individuals to protect themselves from a hazard.

Protection motivation theory (PMT) has quite often been applied to the natural hazards and public health sectors (Floyd et al., 2000; Maddux and Rogers, 1983). According to PMT, people can be motivated to engage in desirable behaviours to avoid risks (Floyd et al., 2000). The theory consists of two sequential underlying cognitive mediating processes, the threat-appraisal process and the coping-appraisal process, whereas assessments of threats (perceived risk severity, vulnerability, and benefits) and coping factors (self-efficacy, response efficacy, and costs) combine to form a motivation in individuals to adopt protective

behaviours. Several studies have applied the PMT to flood risk and analysed its influence on people's risk reduction behaviour. Grothmann and Reusswig (2006) identified that both threat and coping appraisal determined people's adaptation behaviours to increased flood risk in the past (e.g., construction of structural measures, acquisition of protection devices). They conclude that practitioners should communicate not only the flood risk and its impacts, but also the possibility, effectiveness and cost of risk reduction behaviours. Zaalberg et al. (2009) found a significant effect of response efficacy (defined as the degree to

which risk reduction behaviours are perceived to be effective at reducing a particular risk) on behavioural intention, but no effect of self-efficacy (defined as the belief that one is capable or not in undertaking risk reduction measures) on behaviour. Finally, Poussin et al. (2014) found that while threat appraisals had only a small effect on people's intentions to implement flood risk mitigation measures, coping appraisal had a more important influence.

However, PMT does not consider that the public is heterogeneous and may be differently affected by variables of vulnerability,

severity and ability to reduce the risk (Martin et al., 2007). A parallel stream of research in decision stage theories has been used to examine behaviour change based on the assumption that a set of variables, such as from PMT, will influence different people in different ways (Horwath, 1999). The Trans-Theoretical Model (TTM) specifies an ordered set of categories into which people are classified (Martin et al., 2007). Based on this categorization, one can identify the factors (e.g. response efficacy) that can explain how to effectively communicate with each sub-group (i.e. in a specific decision-stage) (Weinstein,

1998). The three decision stages which are frequently used in empirical studies are pre-contemplation, contemplation, and action. The baseline premise is that people can be distinguished based on those who have not yet decided to change their behaviour (pre-contemplation), those who have (contemplation) and those already performing the new behaviour (action) (Martin et al., 2007). Thus, TTM stages are consequential and can change through time.

In consequence, an integrated PMT-TTM model offers the possibility to investigate risk information and perception

dimensions, and stage readiness for action, thus allowing to predict individuals' motivation to act upon the hazard. The PMT-TTM model was introduced by Block and Keller (1998) in the health related context and more recently applied in the natural





hazard context. In these three studies, respondents were grouped in different decision stages based on the amount of suggested protective measures that have already been undertaken. Martin et al. (2007) investigated the role of motivation, decision stages of risk readiness, and subjective knowledge on the number of risk-mitigating actions undertaken by homeowners living in areas exposed to high wildfire risks in the US. Independent of knowledge and decision stage, they found that all PMT variables

were negatively correlated with taking risk reduction behaviours. The authors did ignore the two PMT variables costs and benefits, as well as other control variables. Bočkarjova et al. (2009) analysed the cognitive perception of flood risk and the readiness of people to undertake protective actions in the Netherlands. Overall, they showed a significant influence of response-efficacy and costs on risk reduction measures. The authors used an extended model which also considered other variables and found an influence of subjective knowledge, trust in government, and experience on behaviours. Gebrehiwot and Veen (2015)

applied the PMT-TTM framework to areas of drought risk in the developing countries cultural context (i.e. small-holder farmers in Ethiopia). For the whole sample, they found that vulnerability, severity, self- and response-efficacy are positively and costs negatively correlated with action-taking.

However, it cannot be ignored that there is ambiguity in their findings. The studies do not agree with respect to which PMT variable(s) (vulnerability, severity, benefits, response efficacy, self-efficacy, and costs) influence behavioural intention in the

different decision stages. Moreover, differences also exist in their methodological set-up, and their models differ (e.g. including costs and benefits as PMT variables or not). These differences could have an important effect as, for example, Wachinger et al. (2013) highlighted that perceived benefits and costs can influence individuals' decisions to act upon a risk or not. In addition, Bubeck et al. (2012) argument that a possible feedback from previously adopted protective measures could change risk perceptions. This is in line with findings highlighting that individual expectations of government responses shape the public's

actions in response to floods and longer-term adaptation (Adger et al., 2013; Chamlee-Wright and Storr, 2010). Public adaptation and damage compensation can lead to reduced private protective behaviour, either because citizens view flood mitigation as a government responsibility (Bichard and Kazmierczak, 2012; Botzen et al., 2009; Box et al., 2013; Grothmann and Reusswig, 2006) or because the public adaptation reduced their perception of risk and costs (Raschky and Weck-Hannemann, 2007). Finally, other variables that have been added to the PMT framework in past studies and which appear to

be influential in households' risk-reduction decisions are knowledge (Martin et al., 2007), flood experience (Poussin et al., 2014; Siegrist and Gutscher, 2006, 2008), public assistance (Hanger et al., 2018), and socio-demographic characteristics (e.g., Poussin et al., 2014).

In the research presented in this paper we consider the critics to the PMT-TTM framework and include some variables that previous studies did not consider. We start from the assumption that the designated shift of responsibility on flood protection

from the public to the private sector implies the need to explore the current state of readiness of the public to underake protective action. Moreover there is a need to identify good practices to inform about the adoption of risk reduction behaviours. We use a framework that combines PMT with a model inspired by TTM to identify the factors that trigger individuals to



protect themselves from a hazard, while acknowledging different motivations among a heterogeneous public (see Sect. 3 and Fig. 3). Thus, the main research question is the following: Do different PMT variables (vulnerability, severity, benefits, self-efficacy, response efficacy, costs) affect people's behaviour to undertake risk reduction measures and if so, is it depending on their risk reduction stage?

**3 Case study and research design**

The selected case study is the municipality of Negrar in the Veneto region, north of Verona, on the foothills of the Italian Alps (Fig. 1). The municipality ranges from 70 to 860 m a.s.l. and is drained by the Progno di Negrar and Progno di Novare river systems, both draining to the Adige river, which is the main river basin in the Eastern Italian Alps. Whereas the northern part of the municipality is rather hilly-mountainous, the southern part is characterized by a valley that gets wider and actually turns

into a floodplain. The structure of the municipality is further characterized by several small urban conglomerates (called *frazioni*). The total population of Negrar has steadily increased over the last decades to reach a total of 17093 inhabitants in 2018. As the city is located in a well-known wine area in Italy, the valley is intensively cultivated. The data for this research has been collected in the *frazione* Arbizzano-Santa Maria (Fig. 2, ca. 4000 inhabitants). Arbizzano is located in the floodplain area in the southern part of the municipality and was especially affected by the floods. In the last 80 years, only one flood

event occurred in the summer of 1935 and generated significant damages. The flood risk maps that are used to do the flood risk assessment in the area (called *Piano di Assetto Idrogeologico dell'Adige*) do not indicate any hydrological risk.

Figure 1. Map of Italy with a zoom into the Veneto region. The three major cities Venice, Verona and Padua, as well as the study area, the municipality of Negrar, are shown on the topographic map of Veneto.

On September 1, 2018, the people living in the area experienced a severe flash flood, which impacted the upper part of Progno di Novare river basin, with a drainage area of 2 km$^2$ (outflow at the wine farm known as *Case* Bertani, see Fig. 2). With a cumulated rain of more than 180 mm in less than three hours, the event caused a very quick flood response, with a flood peak around 20 m$^3$/s. Accumulated rain and flood peak are characterised by a return period far larger than 100 year. Hydrological

data (rain and flood peak estimates) were collected and examined during a post flood survey organised on October 5, 2018, by using the methods described by (Amponsah et al., 2018). The collected data include fine resolution weather radar data, which permitted to run a flood simulation to cross-check the internal consistency of rain and peak flood estimates. The event caused inundations and extensive material damage, with around 10 million € damages and 3000 affected people, mostly in the *frazione* of Arbizzano. Short lead times hindered the deployment of preventive measures. After the event, the Veneto Region funded

recovery measures with the implementation of a flood diversion system which aims to divert flood volumes from Progno di Novare to Progno di Negrar. Household protection measures were not publicly funded with this initiative; private insurance schemes are used in a very limited way in the area.



Figure 2. Map of the Progno di Novare (catchment in orange) with its outflow at *Case* Bertani and of Arbizzano (in grey), where most of the flash flood damages were reported.

Techniques of data collection included: exploration of data from existing sources, such as census data and provincial archives; interviews with local authorities to better understand the resident´s risk reduction behaviours; a questionnaire survey carried out interviewing face-to-face a total of 146 residents in Arbizzano-Santa Maria. With regard to our sampling procedure, a context-specific methodology was required to select the respondents, due to the spatial characteristics of flood hazard in the study site. The sample was drawn as to include quotas of respondents selected according to the variables age and gender, so

that it reflects the demographic structure of the local population. Data on the demographics was provided by the municipal administration. The municipality also provided contact persons that were affected by the flood in Arbizzano. Using snowball sampling, these people then provided us with further contacts. However, we also randomly selected households in the most affected areas in Arbizzano. This procedure was necessary to avoid the exclusion or under-representation of precisely those residents we were particularly interested in, i.e. those living in the riskiest areas. The unit of analysis was the individual and

interviewees were instructed to contact only one person per household. The interviewers read out the 28 questions included in the survey and noted the answers of the participants, which were only given a table with different risk reduction behaviours to facilitate answering some questions (see Table 1).

The data collection lasted 2 weeks from 18th of February to 1st of March 2019. The survey was approved by the local authorities

and the data was evaluated anonymously. Participants received no incentive to complete the survey, which took them on average about 30 minutes. Of the 146 residents, 17 people did not own the place they lived in and thus were excluded from data analysis. Additional 5 people were excluded during the data analysis as they did not answer some questions, leaving 124 people. Respondents ranged in age from 20 to 89 years (M = 53.38, SD = 17.99), and n = 77 (52.7 %) were female. The majority of respondents completed high school (44.5 %, n = 65), followed by college or university education (29.5 %, n = 43),

with 5.5 % indicating at least some compulsory education. Compared to the average population in Negrar, the sample was slightly older than the average (M = 44.7 years), similar for the gender ratio of female (50.4 %) and male (49.6 %). Concerning education, people in the sample had similar levels compared to the population in the whole Veneto region (slightly more respondents with high school or university degree balanced by more respondents with only completed middle school).

The questionnaire was prepared on the basis of a review of the existing literature and the results of the documentary analysis, focus groups, semi-structured interviews and existing surveys (Bočkarjova et al., 2009; Hanger et al., 2018; Martin et al., 2007; Poussin et al., 2014). The responses to most of the questions were constructed as 5-point Likert scales, where 1 represented the minimum value and 5 the maximum one. The dependent variable was a composite behavioural measure. It was combined based on an average of eight most common measures that homeowners have undertaken or will undertake to protect their





property from floods. Table 1 provides an overview of the risk reduction behaviours. These behaviours were identified by Poussin et al. (2015) and refined based on the preliminary interviews with authorities and local experts. These interviews also influenced how we operationalized the TTM. Instead of focussing on the decision stages 'pre-contemplative', 'contemplative' and 'action' and the total amount of behaviours undertaken (Bočkarjova et al., 2009; Hanger et al., 2018; Martin et al., 2007;

Poussin et al., 2014), we developed different risk reduction stages that focussed on the quality of behaviours planned or undertaken. We distinguished between three types of protective measures: structural, avoidance and emergency (see Table 1 and Fig. 3). Structural measures include anti-backflow valves, special installation of electric and heating systems. Avoidance measures include keeping personal valuables/documents/expensive appliances above expected flood levels.  Emergency preparedness include the use of mobile barriers and the presence of an emergency plan. Each risk reduction behaviour was

measured using a 5-point scale: 1=will never adopt the measure, 2=will maybe adopt in the long run, 3= will adopt in the near future 4= already done after the flood, 5= already done before the flood, 0=don't know. Based on their highest on average scores, the respondents were divided into one of three risk reduction stages. Even though this different operalization limited the comparability of the studies, we believed that a categorization based on the quality of measures ensures a more effective targeting of communication strategies.

Table 1. Overview of risk reduction behaviours.

The independent variables drew on PMT focussing on threat appraisal (vulnerability, severity, benefits) and coping appraisal (self-efficacy, response efficacy, costs) (see Fig. 3). These 6 variables were computed based on different measures using 5-

point-Likert-scales. We measured vulnerability by asking how much people considered floods as a threat for their house and themselves, and severity by asking what damages they would expect from a future flood of similar severity than the event in September 2018. Self-efficacy was measured by asking how well-prepared people felt to face a future flood, similar to the one experienced in 2018. Response efficacy was measured based on the average score of three items – structural, avoidance and emergency: people had to indicate how effective they thought the three items to be at helping to reduce the risk of floods. Costs

were measured based on the time spend to implement structural, avoidance and emergency measures (three items) and the related financial costs for each of the three types of measures. Finally, benefits were measured based on the average of two items; an extrinsic reward 'I would like to take extra precautions against flooding if I am rewarded or assisted by the government' and an intrinsic reward 'Taking extra risk reduction measures against flooding is a priority'. In addition, we also measured subjective knowledge, trust in government, flood experience, public assistance, having a flood insurance and socio-

demographic variables (age, gender, income, ownership and education). We performed the data analysis using IBM SPSS software, version 23, used for statistical analysis in social science. Several regression analyses were conducted to study the effect of PMT variables in each group and overall. Other covariables were also included in more extended models. Moreover, one-way analyses of variances were conducted to study the effect of socio-demographics on the PMT variables.



Figure 3. Dynamic protection motivation framework that builds on the Protection Motivation Theory (PMT) and the Trans-Theoretical Model (TTM). The three darts highlight that in each group, people are influenced by different PMT variables to engage in risk reduction behaviours.

## 4 Results

We first analysed respondents' answers concerning risk reduction measures. As expected, the adoption of these behaviours is quite low. Figure 4 shows the results revealing that respondents generally indicated higher likelihoods of undertaking avoidance actions (A1-A3, M=2.06) than structural measures (S1-S3, M=1.41) and emergency preparedness measures (E1-E2, M=1.56). Avoidance actions include storing valuables, documents or expensive appliances above expected flood levels. 24 % of all homeowners had already stored valuables above flood levels before the flood in the year 2018 (and since then an
additional 19.2 %), 34.9 % store valuable appliances above flood zones and 15.7 % have an adaptive use of basement (e.g. no sleeping rooms). A minority of people have installed special heating or electricity installations (19.8 %), anti-backflow valves (17,1 %) or use water resistant materials (7.6 %). Likewise, for the emergency measures, less than 20 % have had mobile barriers (such as sandbags) available at home and only 11 % have had an emergency plan. However overall, it is important to highlight that the majority of people never implemented nor is planning to implement any of the measures in the future.

Figure 4. Likert-scale results indicating the likelihood (in %) of undertaking respective risk reduction behaviours.

Based on the eight risk reduction measures, we computed the composite behavioural measure from 1 (will never do) to 4 (already done, before or after the flood). Thus, the higher the score, the higher the stated behaviour to undertake protective
action. Following the respondents' highest on average scores on each type of behaviour (structural, avoidance and emergency), we further divided the respondents in our sample into three risk reduction stages: structural, avoidance, and emergency. Thus, those who have already undertaken more structural measures (or are most likely to do so), compared to the avoidance and emergency measures, were put into the structural group. Those who have adopted most, or were most likely to adopt avoidance measures, were grouped in the avoidance group and participants with highest scores on emergency preparedness measures
were integrated into the emergency group. Thus, the 124 participants were divided into the three groups as follows: 26 in the emergency preparedness group, 53 in the avoidance group and 45 in the structural group.

We continue by analysing the influence of different PMT variables on undertaking risk reduction measures in each risk reduction stage. Table 2 shows the effects of the variables vulnerability, severity, self-efficacy, response-efficacy, benefits and
costs on the composite risk reduction behaviour measure for each stage. The associated descriptive statistics are provided in Table 3. The results for the first regression for the emergency group show that the only significant predictors of risk reduction behaviour are perceived severity of damage and costs. The higher the perceived damage, respectively the lower the perceived



costs, the more likely it is that people will undertake emergency measures. In the avoidance stage, vulnerability and self-efficacy are the most important predictors. The more people considered floods to be a threat and the more they believe to be well prepared to face a flood, the more likely they are to adopt avoidance behaviours. The third regression highlights that the significant predictors for people in the structural group are vulnerability, severity, response-efficacy and costs. People at this

stage are more likely to be motivated to engage in structural risk reduction behaviours by various factors: high perceived vulnerability, response-efficacy, and weak perceptions of hazard severity and costs. The overall model indicates that, independent of risk reduction stage, coping factors (self and response efficacy) are more important than the assessment of threats in motivating people to engage in risk reduction behaviours. At the same time higher costs, both in terms of time to implement a measure and money to finance it, demotivate people to engage in risk-reduction behaviours. Moreover, costs

seem to be more important than benefits/rewards (i.e. taking extra precautions against flooding because it is a priority for the household or only if rewarded by government). Overall, these findings highlight that the three groups of property owners can be differentiated based on their stage readiness, which has important consequences on effectively communicating flood risk (see Sect. 5).

Table 2. Beta coefficients and p-values for risk reduction behaviours regressed on vulnerability, severity, self-efficacy, response-efficacy, benefits and costs.

Table 3. Descriptive statistics of variables used in the dynamic protection motivation framework.

Interestingly, the additional variables (trust, experience, knowledge, public assistance and socio-demographics) do not influence people's behaviours to act upon the risk in the extended model. Even though we find some differences with respect to PMT variables, this does not translate into a significant difference on the composite risk reduction behavioural measure. Some interesting results in this respect are that women scored significantly higher than men on the threat factors (vulnerability ($F=3.98$, $p=.048$) and severity ($F=9.43$, $p=.003$)) and on response-efficacy ($F=6.44$, $p=0.013$), as well as lower on self-efficacy

($F=7.0$, $p=0.01$).

Furthermore, we also identify specific trends that can be found for each PMT variable across the three risk reductions stages. The colours in Table 2 shall help to visually summarize the key trends. For severity of expected damages, we can observe a linear trend. Severity is positively correlated with taking protective action in the group relying on emergency measures and

negatively correlated in the group relying on structural measures. Vulnerability shows the exact opposite pattern. Negative (but insignificant) in the group relying on emergency measures, it becomes positive and is a significant motivator for people to undertake avoidance or structural measures. Response efficacy shows a similar trend and becomes positively correlated with risk reduction behaviour in the groups relying on structural measures. Self-efficacy plays a critical role in the avoidance group




and is less important in the other groups. Overall, costs have a negative impact, but are of special importance for people relying on structural measures.

## 5 Discussion

In order to better understand what motivates people to protect against flood risk we developed a dynamic protection motivation
framework. We tested it in a municipality in Northern Italy that experienced severe flash flooding in 2018, shortly before the survey was conducted. The framework proved to be useful for assessing people's perception of flood risk and their readiness to undertake risk reduction behaviours. Our findings add to the other studies in the natural hazard field (Bočkarjova et al., 2009; Gebrehiwot and Veen, 2015; Martin et al., 2007), in which people were grouped based on their risk reduction behaviours. Concerning the overall results (i.e. independent of group membership), we confirm most of the prior findings. We support
Bočkarjova et al. (2009) and Gebrehiwot and Veen (2015) who also found a negative effect of costs on risk reduction behaviours. Alike both studies, we also show that people with strong perceptions of response-efficacy have a higher likelihood in taking risk reduction actions. Furthermore, we support Gebrehiwot and Veen, (2015), who showed a positive effect of self-efficacy. However, we do not find a significant overall effect of vulnerability as the other two aforementioned studies did, which could be due to the different operalization of the variable.

Our results suggest that people are motivated by different factors in prompting risk reduction behaviours based on their risk reduction stage. We show that PMT variables influence different groups of people in different ways. Structural and avoidance groups are more likely to be motivated by increased perceptions of vulnerability and response efficacy and are less worried by expected flood damages compared to emergency group. Self-efficacy also plays an important role in the avoidance group,
whereas perceived severity of a hazard is an important factor in the emergency group. Through all the groups, costs have a significant negative impact on risk reduction behaviours, whereas benefits do not matter at all. Independent of group membership, we confirm Poussin et al. (2014) and find that coping appraisal had a more important influence then threat appraisal. High perceptions of self-efficacy and response-efficacy significantly increase the likelihood to engage in risk reduction behaviour. We also support Wachinger et al. (2013) and show that costs are the biggest impediment towards action-
taking. Beside costs, other reasons may also explain the low adoption of the risk reduction behaviours. The majority of people indicated that they did not think that a flood similar to 2018 could have happened and many also believed that it will not happen again anytime soon. Thus, we speculate that the non-existing flood history (e.g. low perceived risk exposure) could explain the low adoption of risk reduction behaviours before and also the limited willingness to implement measures in the aftermath of the disaster.


Beside the identification of factors prompting risk reduction behaviours, we also identify specific trends which help to explain the current readiness of individuals to undertake protective actions. The data shows that severity plays an important role in



motivating people to act upon emergency behaviours, but as people adopt more significant protections (such as anti-backflow valves or special installations of heating/electric system), severity acts more as an inhibitor. It seems that these people feel more protected and thus do less bother about the severity of a hazard and its expected damages. For response-efficacy the opposite is true. This indicates that, as the measures become more sophisticated, they are also perceived to be more effective

in reducing or even preventing flooding. For self-efficacy, there is no trend as it is only significant in the avoidance group. This highlights that people feel ready in undertaking avoidance measures on their own, but less so for the structural measures, and in consequence they are not motivated by self-efficacy to engage in structural risk reduction behaviours. In this respect, it is surprising to see that self-efficacy is not significant in the emergency group, as these measures (e.g. storing sandbags, having an emergency plan) could be more easily performed (like the avoidance measures). We speculate that there could be a trade-

off between experience (either being hit hard by the flood or not), efficacy of measures to prevent damage and self-efficacy which could explain this observation. As the measures in the structural group are the most cost intensely, costs act as an important barrier here. This could also explain why costs are no such significant barrier in the avoidance group. These measures (e.g. store valuables or boiler above the expected flood level) do not require much effort to be implemented, both in terms of financing and time. Intrinsic or extrinsic rewards (i.e. taking extra precautions against flooding because it is a priority for the

household or only if rewarded by government) do not seem to be of any relevance in our sample. Concerning the extrinsic reward, this could be explained by the non-existing public support after the flood and the, in general, low trust levels in the government. Concerning the intrinsic reward, this could be an indication that people in mountainous (as opposite to urban) areas feel less the pressure of society to show themselves in the best possible light.

We also explored a number of possible covariates, which could explain different levels of behaviour to act upon the risk. We do not find any significant influence of these variables on the composite risk reduction behavioural measure, but show an effect of gender on all PMT variables. Generally, women are more likely to engage in protective behaviour compared to men (e.g., Lazo et al., 2015; Morss et al., 2016). We find that women are more threatened by the hazard and its expected damages, and think measures to be more efficient than male, while they feel less prepared to face a hazard. As this does not translate into

differences in risk reduction behaviours, it could indicate that even though women prefer to be more protected, they do not have the decision power in the household to implement the measures.

## 6 Conclusions

Our findings also draw implications for influencing individuals in performing desirable adaptive behaviours. Property owners in flood risk areas should not be considered as a same homogeneous community that can be targeted with a single

communication strategy. Thus, communication strategies should not be applied homogenously across large areas, but instead should be tailored to the individual motivations to act upon a risk or not in order to be effective. For people that are most likely to undertake emergency measures, increasing the perceptions of severity rather than focusing on perceptions of vulnerability



will lead to greater risk reduction behaviours. The communication strategy should further try to decrease perceptions of costs and to describe benefits of the risk reduction measures. This can be done by providing extra information about flood risk and especially its consequences to these people. Moreover, storytelling can be also used as a powerful communication tool: positive and negative experiences of fellow residents that respectively adopted and didn´t adopt the measures can help to drive

behavioural change. For the people in the avoidance group, the communication strategy should focus on the information concerning flood likelihood in the residence area and on the self-efficacy of a number of proposed flood risk reducing measures. Folders, web platforms or information events could help to raise awareness and increase flood preparedness.

In the structural group, the perceptions of costs associated with taking protective action (such as time, effort and money) should be decreased. Thus, communication should stress the efficacy of structural measures, in relation to the costs and give clear

explanation of the effectiveness of the proposed measures. In addition, the communication should focus on information concerning flood likelihood in the area of residence and not emphasize hazard severity as it only seems to decrease the likelihood to act upon the risk. Finally, public assistance from the government, be it financial aid for financing flood protection (e.g. subsidies, loans, tax reliefs, etc.) or material support (e.g. distribution of sandbags), could help overcome costs, which are the biggest impediment, and outweigh the balance in favour of acting. Ultimately, these new insights should help to develop

better targeted flood risk communication strategies to individuals that emphasize different motivating aspects and foci for each risk reduction stage. An enabling condition to do so will be the integration of risk perception and preparedness surveys in local flood risk management practices. These surveys should be included in longitudinal studies, which may also help to better understand the driving factors of risk reduction behaviours through time. Especially in the areas at highest risk, short surveys could be a valuable (cost and time efficient) instrument to find out at what risk reduction stage people are. If survey information

is not available, practitioners may decide a-priori the desired or most likely risk reduction measure (emergency, avoidance or structural) for a community or an area and target their communication accordingly. This means that they could then rely on the specific PMT variables identified in this study to motivate people to undertake the selected measures.

### Data availability

A DOI was generated and reserved for the data: 10.17632/7hwtd84g9s.2. However, please note that the submission is currently under review and that the uploaded digital object will not be visible to external users until approved by staff.

### Author contribution

PW designed the framework and developed the research question. PW also conducted the data analysis and wrote most of the

manuscript. EM did the data collection. AS helped designing the survey and wrote some parts of the manuscript. MB helped carrying out the field survey and provided local information on the case study. GDB and AP revised the draft and provided comments and inputs.




## Competing interests

The authors declare that they have no conflict of interest.

## Acknowledgements

We thank the municipality of Negrar for their valuable inputs. We also thank the master students from the University of Padova that helped to collect the data.

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





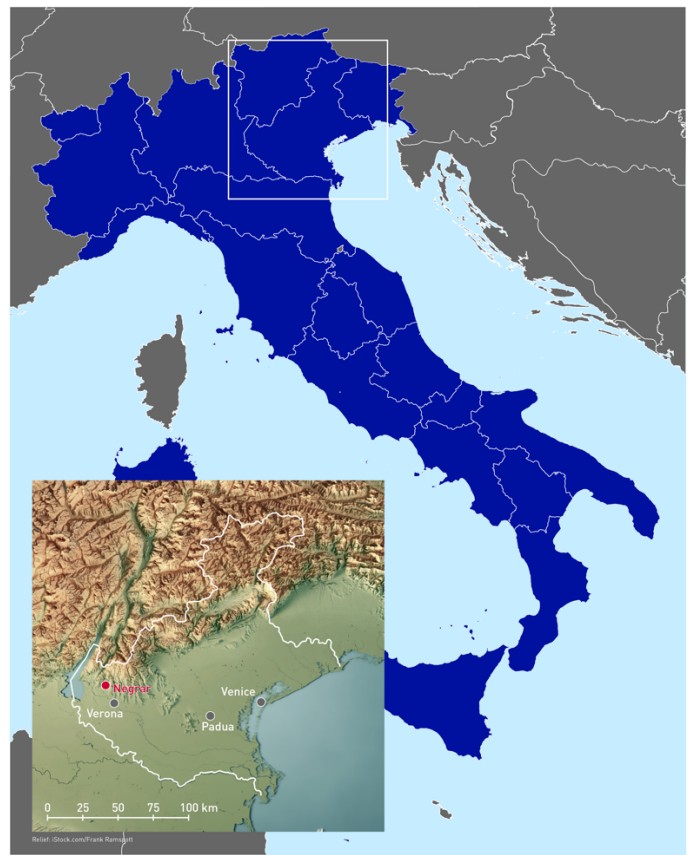

**Figure 1: Map of Italy with a zoom into the Veneto region. The three major cities Venice, Verona and Padua, as well as the study area, the municipality of Negrar, are shown on the topographic map of Veneto.**



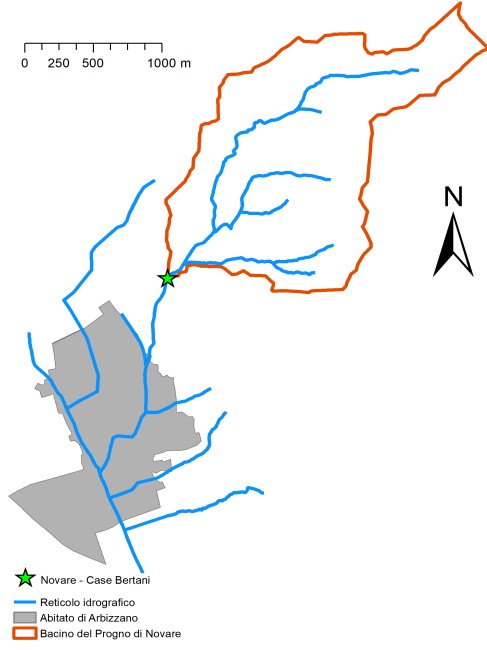

**Figure 2: Map of the Progno di Novare (catchment in orange) with its outflow at *Case* Bertani and of Arbizzano (in grey), where most of the flash flood damages were reported.**

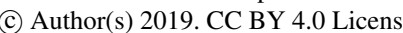



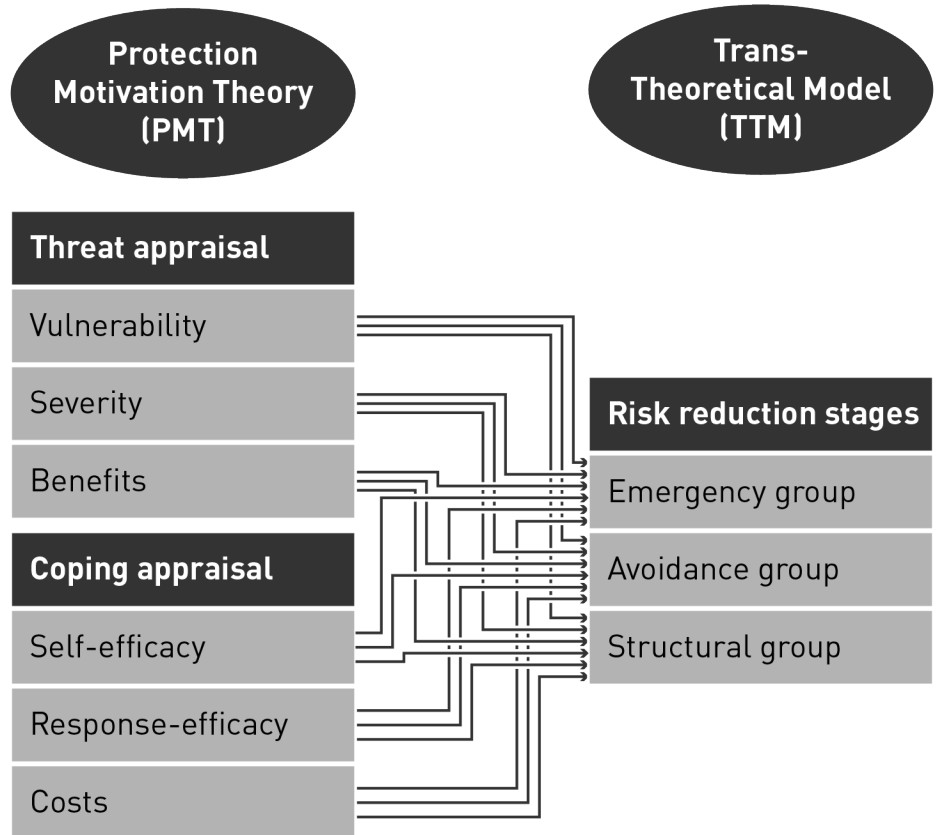

**Figure 3: Dynamic protection motivation framework that builds on the Protection Motivation Theory (PMT) and the Trans-Theoretical Model (TTM). The three darts highlight that in each group, people are influenced by different PMT variables to engage in risk reduction behaviours.**




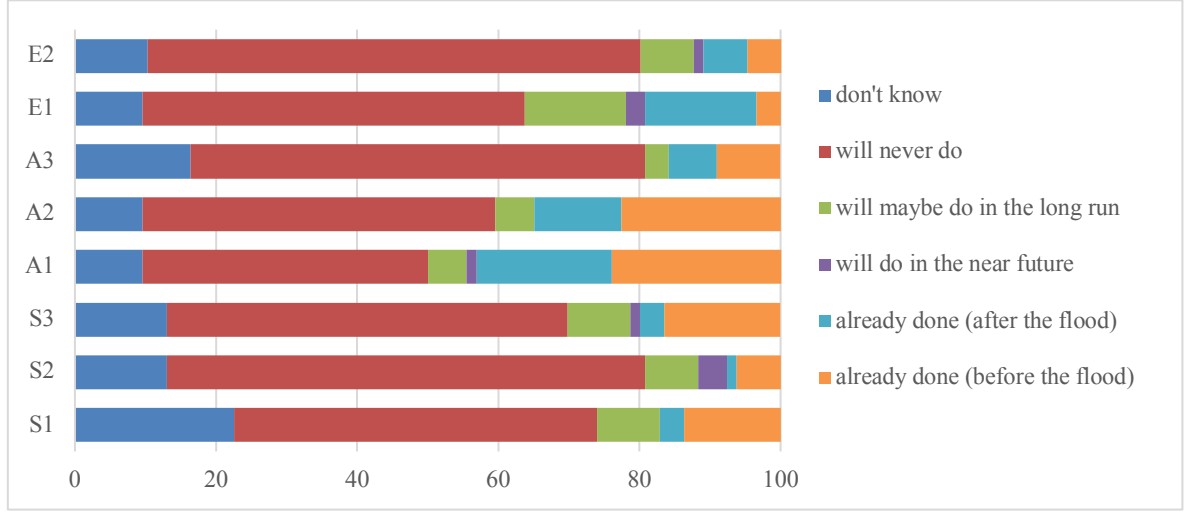

**Figure 4: Likert-scale results indicating the likelihood (in %) of undertaking respective risk reduction behaviours.**



**Table 1: Overview of risk reduction behaviours.**

| Type of behaviour | Risk reduction behaviour | Abbreviation (referring to Fig. 4) |
|---|---|---|
| Structural measures | Anti-backflow valves | S1 |
| | Ground floor and walls made of water-resistant materials | S2 |
| | Special installation (e.g. higher up) of heating and electric system | S3 |
| Avoidance measures | Keeping personal valuables and documents out of flood-prone areas of the house | A1 |
| | Keeping expensive appliances (washing machine, boiler etc.) above expected flood levels | A2 |
| | Adapted use of basement and ground floor | A3 |
| Emergency preparedness | Mobile barriers (e.g. metal/wood shields) available | E1 |
| | Emergency plan for household in case of floods (e.g. where to go, what to take with me) | E2 |





**Table 2: Beta coefficients and p-values for risk reduction behaviours regressed on vulnerability, severity, self-efficacy, response-efficacy, benefits and costs.**

|  | Risk Reduction Stage Readiness | | | |
|---|---|---|---|---|
|  | Emergency | Avoidance | Structural | Overall |
| Vulnerability | -.477 | .609* | .664* | .195 |
| Severity | .702* | -.362 | -.672* | -.023 |
| Self-efficacy | -.223 | .922** | .181 | .325* |
| Response-efficacy | -.241 | .253 | .456* | .280* |
| Benefits | .021 | -.187 | -.008 | -.147 |
| Costs | -1.067* | -.154 | -.741** | -.541** |
| F-ratio | 4.331* | 6.701** | 5.194** | 6.228** |
| $R^2$ | 0.79 | 0.73 | 0.66 | 0.42 |

*Note*: $R^2$ represents the amount of variance in the outcome explained by the model relative to the total variance. *$p$ < 0.05 and **$p$ < 0.001. To highlight trends in the three risk reduction stages, green colours indicate significant positive correlations and orange colours indicate significant negative correlations.





**Table 3: Descriptive statistics of variables used in the dynamic protection motivation framework.**

|  | Emergency | | Avoidance | | Structural | | Overall | |
|---|---|---|---|---|---|---|---|---|
|  | Mean | St. Dev. | Mean | St. Dev. | Mean | St. Dev. | Mean | St. Dev. |
| Composite behavioural measure | 1.57 | 0.48 | 1.68 | 0.49 | 1.65 | 0.66 | 1.64 | 0.55 |
| Vulnerability | 3.46 | 1.37 | 3.53 | 1.46 | 2.84 | 1.31 | 3.3 | 1.44 |
| Severity | 2.42 | 1.42 | 3.43 | 1.23 | 2.71 | 1.12 | 3.16 | 1.29 |
| Self-efficacy | 2.42 | 1.21 | 2.33 | 1.16 | 2.28 | 1.16 | 2.31 | 1.15 |
| Response-efficacy | 3.57 | 1.14 | 3.56 | 1.11 | 3.63 | 1.10 | 3.59 | 1.10 |
| Benefits | 4.38 | 0.95 | 4.02 | 1.15 | 3.86 | 1.05 | 4.03 | 1.10 |
| Costs | 3.23 | 0.71 | 3.24 | 0.47 | 3.45 | 0.69 | 3.31 | 0.60 |

