# Peer review of "A flood risk oriented dynamic protection motivation framework to explain risk reduction behaviours"

_Natural Hazards and Earth System Sciences, 2019_

## Referee Comment (RC1) · Anonymous Referee #1 · 4 Aug 2019

This is a well-written manuscript, providing adequate details and good scientific quality. The results and conclusions are clear, concise and focused enough. The manuscript is discussing the motivations behind the adoption of protective actions by the public with the novel idea of combining PMT with TTM to deal with the non-homogeneity of population. The manuscript is worth publishing as it clearly adds to the knowledge in the field of risk perception and protective behaviors, in the wider topic of flood risk mitigation. I recommend accepting the manuscript with minor revisions, provided in the following list: - Page 2: Line 23-25: It is important to note that in the region and possibly elsewhere, it has been found that laymen think of flooding as a man-made or man-caused phenomenon rather than a natural one (Lara et al. 2010 and Diakakis

et al. 2018). Lara A., D. Sauri, A. Ribas, D. Pavon, Social perceptions of floods and flood management in a Mediterranean area (Costa Brava, Spain), Nat. Hazards Earth Syst. Sci. 10 (2010) 2081–2091. Diakakis M., Priskos G., Skordoulis M (2018) Public perception of flood risk in flash flood-prone areas of Eastern Mediterranean: the case of Attica Region in Greece. International Journal of Disaster Risk Reduction, 28, 404–413

- Page 3: lines 15-17: Although it can be understood, there is a logical leap, that may be hard for the reader to understand. Why this categorization allows us to identify the key issues to improve communication with people in each stage? It is not well connected with the previous phrase. I would recommend to describe it in a bit more detail as it is a very crucial part of the manuscript (i.e. setting the objectives).

- Page 3: lines 25-27: This seems to be an important argument, to make your case (to build the problem presentation). It is in the right direction. So, I would recommend adding more literature here. I recommend literature that states that there is no link between risk perception and actual adoption of mitigation measures. K. Takao K, T. Motoyoshi, T. Sato, T. Fukuzono, Factors determining residents' preparedness for floods in modern megalopolises: the case of the Tokai flood disaster in Japan, J. Risk Res. 7 (2004) 775–787 Kreibich, H. A.H. Thieken, T. Petrow, M. Müller, B. Merz, Flood loss reduction of private households due to building precautionary measures - lessons learned from the Elbe flood in August 2002, Nat. Hazards Earth Syst. Sci. 5 (2005) 117–126. Siegrist, M., H. Gutscher, Flooding risks: a comparison of lay people's perceptions and expert's assessments in Switzerland, Risk Anal. 26 (2006) 971–979. Diakakis M., Priskos G., Skordoulis M (2018) Public perception of flood risk in flash flood-prone areas of Eastern Mediterranean: the case of Attica Region in Greece. International Journal of Disaster Risk Reduction, 28, 404–413

- Page 6: line 26: I think the parentheses should not include the author. The authors should keep the parenthesis only for the year in Amponsah et al. 2018 - Page 6: line 28: the word damage repeats. Please reword. - Page 6 line 30-32: The diversion may

have played a role in people's perception. There have been cases where inhabitants of areas that see public flood protection works are influenced in their thinking to a degree. There could be a question on the survey on that or to measure that. If it there hasn't been a question, maybe it could be acknowledged by the authors as a limitation of the current research, stating that the current survey did not examine the influences of significant flood protection measures separately. - Page 8: line 30: family status (especially the presence of young children) has been correlated with protective behaviors and risk perception in previous works. I assume it was not surveyed. Given the numerous other factors that were examined the manuscript has a lot of merit and value for publishing. In future surveys, though I recommend being included. If the family status was surveyed in the present study, then I would like to see how it is correlated. - Figure 4: In the caption, does the word "likelihood" refers to absolute likelihood or reported-likelihood by the respondents of the survey? If it is the latter, then it should probably be revised to "reported-likelihood". At the moment it is a little confusing for the reader - page 10: lines 11-12. Sentences not perfectly clear. Please elaborate further or make sentence simpler. - Page 11: In general in the discussion section, it could be of value to mention in more detail what did you find regarding "experience" of respondents. The literature clearly shows that perception is correlated with experience and that the latter is an important factor. Currently is not discussed adequately. - Page 11: line 29: The way you describe the findings in the last paragraph, what comes in mind is a simple concept of "risk personalization" or the phrase "it won't happen to me". This has been noted in the literature. For example Gissing et al. 2016, clearly states that people ignored warnings and went around barricades and drove into flooded areas, even though protective behavior would mean something very simple such as a d-tour. I think this concept has to be acknowledged in the discussion to a limited extent, in the sense that not everything can be projected or explained. There will always be a factor such as the failure to "personalize risk", even though this manuscript provides excellent ground to reduce this uncertainty of predicting behavior. Gissing, A., Haynes, K., Coates, L., & Keys, C. (2016). Motorist behavior during the 2015 Shoalhaven floods.

[Figure]

Australian Journal of Emergency Management, The, 31(2), 25.

Page 12: line 9-11: Sentence not so clear. Please rephrase to make it more clear.

Page 12: line 17-18: Please state clearly that this is speculation, to avoid the risk of other authors take it as a data-based conclusion and propagate assumptions in their own research.

Page 12: line 23: Previous studies refer to differences in risk perception between male and females. I think a few more remarks on this very subject should be included here.

Page 12: line 26: any references for this claim?

Page 12: line 27: the word "also" is not needed here

---

## Author Comment (AC1) · 3 Sep 2019

**A flood risk oriented dynamic protection motivation framework to explain risk reduction behaviours**

**Response to the anonymous reviewer's comments in the interactive discussion: nhess-2019-120**

| Nbr. | Page/line | Reviewer 1 | Author's response |
|---|---|---|---|
| 1 | / | This is a well-written manuscript, providing adequate details and good scientific quality. The results and conclusions are clear, concise and focused enough. The manuscript is discussing the motivations behind the adoption of protective actions by the public with the novel idea of combining PMT with TTM to deal with the non-homogeneity of population. The manuscript is worth publishing as it clearly adds to the knowledge in the field of risk perception and protective behaviors, in the wider topic of flood risk mitigation. I recommend accepting the manuscript with minor revisions, provided in the following list: | We thank the Reviewer very much for his/her numerous suggestions and comments. We addressed all the comments made that helped to improve the manuscript. |
| 2 | 2/23-25 | It is important to note that in the region and possibly elsewhere, it has been found that laymen think of flooding as a man-made or man-caused phenomenon rather than a natural one (Lara et al. 2010 and Diakakis et al. 2018). | We inserted a sentence and included the literature references: "This may be related to fact that people perceive flooding as a man-made phenomenon rather than a natural one, for example in the Mediterranean region (Diakakis et al., 2018; Lara et al., 2010)." |
| 3 | 3/15-17 | Although it can be understood, there is a logical leap, that may be hard for the reader to understand. Why this categorization allows us to identify the key issues to improve communication with people in each stage? It is not well connected with the previous phrase. I would | We inserted a sentence to better explain the connection between PMT-TTM and our research question: "Combining PMT with TTM will help to deal with the non-homogeneity of the population, who is not equally ready to adopt protective actions, and to better understand the respective motivations in order to reduce some of the uncertainty in predicting flood risk behaviour This framework allows us |

| | | recommend to describe it in a bit more detail as it is a very crucial part of the manuscript (i.e. setting the objectives). | to further identify the key issues to improve communication with people in each stage." |
|---|---|---|---|
| 4 | 3/25-27 | This seems to be an important argument, to make your case (to build the problem presentation). It is in the right direction. So, I would recommend adding more literature here. I recommend literature that states that there is no link between risk perception and actual adoption of mitigation measures. | We included the recommended literature in a couple of sentences: "For example, Siegrist and Gutscher, (2006) found that even though people living in areas with higher levels of designated risk had higher risk perceptions than people living in areas of no flood risk, there was no difference in prevention behaviour between the groups. Other studies that were conducted in flooding areas documented an overall low level of preparedness and observed no relationship between risk perception and preparedness (Diakakis et al., 2018; Kreibich et al., 2005; Takao et al., 2004).» |
| 5 | 6/26 | I think the parentheses should not include the author. The authors should keep the parenthesis only for the year in Amponsah et al. 2018 | Done. |
| 6 | 6/28 | the word damage repeats. | Removed. |
| 7 | 6/30-32 | The diversion may have played a role in people's perception. There have been cases where inhabitants of areas that see public flood protection works are influenced in their thinking to a degree. There could be a question on the survey on that or to measure that. If it there hasn't been a question, maybe it could be acknowledged by the authors as a limitation of the current research, stating that the current survey did not examine the influences of significant flood protection measures separately. | We agree with the Reviewer. In the survey we asked participants how much they agreed with statements whether protection works eliminate the possibility of serious damages or whether they give a sensation of security. However, these questions were not specifically related to the implementation of the flood diversion system. Thus, we inserted a sentence at the end of the manuscript which acknowledges the limitation: "We examined the motivations of people to undertake risk reduction behaviours, but did not analyse the influences of significant flood protection measures (such as the flood diversion system) separately." |
| 8 | 8/30 | family status (especially the presence of young children) has been correlated with protective behaviors and risk perception in previous works. I assume it was not surveyed. Given the | Thanks for this valuable comment. Unfortunately, we did not survey the family status of participants, but will do so in future surveys! |

| | | numerous other factors that were examined the manuscript has a lot of merit and value for publishing. In future surveys, though I recommend being included. If the family status was surveyed in the present study, then I would like to see how it is correlated. | |
|---|---|---|---|
| 9 | Figure 4 | In the caption, does the word "likelihood" refers to absolute likelihood or reported-likelihood by the respondents of the survey? If it is the latter, then it should probably be revised to "reported-likelihood". At the moment it is a little confusing for the reader | We re-worded the caption: "Results indicating the percentage of responses in each category of the Likert-scale for each risk reduction measure." |
| 10 | 10/11-12 | Sentences not perfectly clear. Please elaborate further or make sentence simpler. | We removed the sentence and replaced it by a simpler one. |
| 11 | 11 | In general, in the discussion section, it could be of value to mention in more detail what did you find regarding "experience" of respondents. The literature clearly shows that perception is correlated with experience and that the latter is an important factor. Currently is not discussed adequately. | We found no effect of experience as we stated in the last paragraph of the discussion. We added a bracket "(experience, trust and socio-demographics)" to name the covariables and make it more clear. We also added some sentences on previous research. "Concerning experience with floods and natural hazards, most studies found that risk perceptions and mitigation behaviours correlate with experience (Bubeck et al., 2012), even though some exceptions exist (Takao et al., 2004). Likewise, trust was also found by previous research to have direct and indirect effects on flood preparedness intentions (e.g., Terpstra, 2011)." |
| 12 | 11/29 | The way you describe the findings in the last paragraph, what comes in mind is a simple concept of "risk personalization" or the phrase "it won't happen to me". This has been noted in the literature. For example Gissing et al. 2016, clearly states that people ignored warnings and went around barricades and | We added some sentences here (including the reference) to acknowledge the concept and that not everything can be explained: "Similarly, other research observed that people do not protect themselves even though the protective behaviour would be very easy to adopt. Gissing et al. (2016) observed that motorists ignored warnings and drove into flood waters despite an obvious risk of death. The authors list numerous motivations such as underestimating or not understanding the risk, |

| | | drove into flooded areas, even though protective behavior would mean something very simple such as a d-tour. I think this concept has to be acknowledged in the discussion to a limited extent, in the sense that not everything can be projected or explained. There will always be a factor such as the failure to "personalize risk", even though this manuscript provides excellent ground to reduce this uncertainty of predicting behavior. | feeling invincible and not taking the warning seriously. Hence, we have to acknowledge that each person sees and responds slightly differently to flood risk, and try to reduce the uncertainty of predicting behaviour as much as possible." |
|---|---|---|---|
| 13 | 12/9-11 | Sentence not so clear. Please rephrase to make it more clear. | Indeed the sentence was not clear and we decided to delete it, as the speculation did not add to the understanding of the paragraph. |
| 14 | 12/17-18 | Please state clearly that this is speculation, to avoid the risk of other authors take it as a data-based conclusion and propagate assumptions in their own research. | Done. |
| 15 | 12/23 | Previous studies refer to differences in risk perception between male and females. I think a few more remarks on this very subject should be included here. | We included a sentence and stated some literature: "Previous studies also reported differences in risk perception between men and women. On average, men are found to have lower perception levels of flood risk compared to women (Ho et al., 2008; Lindell and Hwang, 2008), even though (Botzen et al., 2009a) found the opposite relationship." |
| 16 | 12/26 | any references for this claim? | No, we made clear that this is a speculation. |
| 17 | 12/27 | the word "also" is not needed here | Done. |

---

## Referee Comment (RC2) · Anonymous Referee #2 · 3 Nov 2019

General comments

The paper presents a method combining two theories to understand better motivation of people to undertake protection measures against floods. The paper is well written and well analyzed and is of interest for the readers of Natural Hazard and Earth System Sciences (NHESS). The authors argue that their approach, that splits people groups, according to a classification related to the type of protection measures people are ready to undertake, could help targeting better communication about risk reduction.

I believe that the paper could be further improved by providing more details about the context (flood prevention and alert) in the study area. The authors should also dis-

cuss how the combination of the two theories (Protection Motivation Theory – PTT) and Trans-Theoretical Model (TTM) and their implementation, mentioned to be different from previous ones, proved to be relevant and the limits they identified in their approach. They should also discuss how their methodology/results could be adapted to other countries/contexts. The conclusions section should also be rewritten, as its present content belongs more to the discussion section than to the conclusion section. The specific comments listed below should also be addressed before publication.

Therefore, I suggest moderate revision of the paper before publication in NHESS.

Specific comments

1/ p.3, line 14: the authors should precise what they mean by "flood risk". I guess they are referring to the "static" risk of people being affected in their home, as opposed to dynamic flood risk, when people are exposed during their mobility.

2/ P.3, line 18: when the authors write "motivations to protect themselves", do they mean to protect their property, not their life?

3/ P.4 line 28: I do not understand the last sentence. Consequential – do you mean sequential?

4/ P.5 line 1 "In these three studies.." to which studies does "these" refer to?

5/ P.5 line 30 " of the public to undertake."

6/ P.6 lines 14-16: could the authors provide more information about the flood risk maps. Are they built to map areas at risk of floods of different return periods? How is the information about flood risk communicated to people. This could be an important element for the interpretation of the results of the study and I believe it is important to communicate this information for people from countries where flood risk information could be different. In addition, how do you explain that some people had taken risk reduction measures before the flood if the area was not declared at risk?

[Figure]

7/ P.6 lines 26-27: Is there any warning to people when heavy precipitation is expected in a region? How were the losses compensated if few people subscribe to a insurance policy?

8/ P.6 in the presentation of the case study, could you also indicate the types of houses that were affected by the flood: are they located in old or new parts of the town? Do the house have several floors (it seems to be the case as basement is mentioned p. 9 line 11)?

9/ P.7 lines 16-17: did some people mention other protection measures than the ones that are included in Table 1?

10/ P.7 lines 24-28: it could be easier for the reader to present the percentages in terms of increasing of decreasing level of education. Furthermore, the authors use "compulsory school" and "middle school": it is the same thing.

11/ P.8 lines 1-14: If you are able to affect the different interviewees in the three categories, it means that they are aware that there is a risk and are ready to protect themselves. Are you sure of that? Apparently not as you mention p. 9 line 14 that the majority of people are not ready to implement any protection measure.

12/ P.8 line 20: when you ask people if they consider their house to be at risk of flooding, are you referring to their perception of the risk or to objective information like maps of flood prone areas?

13/ P.8 line 31: You mention the use of regression analysis: does it mean that all the variables were transformed to quantitative values? How did you perform that for "benefits"?

14/ P.9 line 14: you mention that the majority of people has not or did not plan to perform protective measures. How did you classify these people in one of the three categories?

15/ P.11 line 2: Have you any idea of the results of the survey if it has been conducted

before the flood?

16/ P.11 line 22 "more important influence than threat"

17/ P.11 line 26-27: when you performed the survey, had people already recovered from the flood? Did some people were affected on the long term? Did they received any compensation for the damages? Which type of communication would be required so that people remain aware of the risk of flash flooding?

18/ P.12 lines 17-18: the last sentence is not clear.

19/ Discussion: could you add some elements in the discussion explaining how general/particular are the results of your case study? You should also discuss the relevance of your methodology as you mentioned that you applied the PTT-TTM models differently than others.

20/ P.12-13: what is presently in the conclusions, should appear in the discussion and the conclusions should include the main findings of the study, not possible outcome of the study.
* * *

---

## Author Comment (AC2) · 19 Nov 2019

**A flood risk oriented dynamic protection motivation framework to explain risk reduction behaviours**

| Nbr. | Page/ | Reviewer 2                                                                                                                                                                                                                                                                                                                                                                                                                                                                                                                                                                                                                                                                                                                                                                                                                                                                                                                                                                                                                                                                                                                                                                                                                           | Author's response                                                                                                                                                                                                                                                                                                                                                                                      |
|------|-------|--------------------------------------------------------------------------------------------------------------------------------------------------------------------------------------------------------------------------------------------------------------------------------------------------------------------------------------------------------------------------------------------------------------------------------------------------------------------------------------------------------------------------------------------------------------------------------------------------------------------------------------------------------------------------------------------------------------------------------------------------------------------------------------------------------------------------------------------------------------------------------------------------------------------------------------------------------------------------------------------------------------------------------------------------------------------------------------------------------------------------------------------------------------------------------------------------------------------------------------|--------------------------------------------------------------------------------------------------------------------------------------------------------------------------------------------------------------------------------------------------------------------------------------------------------------------------------------------------------------------------------------------------------|
|      | line  | The paper presents a method
combining two theories to
understand better motivation
of people to undertake
protection measures against
floods. The paper is well written
and well analyzed and is of
interest for the readers of
Natural Hazard and Earth System
Sciences (NHESS). The authors
argue that their approach, that
splits people groups,
according to a classification
related to the type of protection
measures people are ready
to undertake, could help
targeting better communication
about risk reduction.
I believe that the paper could be
further improved by providing
more details about the
context (flood prevention and
alert) in the study area. The
authors should also discuss how
the combination of the two
theories (Protection Motivation
Theory – PTT)
and Trans-Theoretical Model
(TTM) and their implementation,
mentioned to be different from
previous ones, proved to be
relevant and the limits they
identified in their
approach. They should also
discuss how their
methodology/results could be
adapted
to other countries/contexts. The
conclusions section should also | We thank the Reviewer very much for
his/her numerous suggestions and
comments. We addressed the specific
comments in the following. In general, we
provide more information about the study
area and discussed in more detail the PMT-
TTM approach and its limits. We
restructured the conclusion and discussed
the generalization of our results for
different countries. |

Response to the anonymous reviewer's comments in the interactive discussion: nhess-2019-120

|   |         | present content belongs more to
the discussion section than to the
conclusion section.                                                                                                                                                                                                                                                                                                                                                                                                                                                                    |                                                                                                                                                                                                                                                                                                                                                                                                                                                                                                                                                                                                                                                                   |
|---|---------|-----------------------------------------------------------------------------------------------------------------------------------------------------------------------------------------------------------------------------------------------------------------------------------------------------------------------------------------------------------------------------------------------------------------------------------------------------------------------------------------------------------------------------------------------------------------|-------------------------------------------------------------------------------------------------------------------------------------------------------------------------------------------------------------------------------------------------------------------------------------------------------------------------------------------------------------------------------------------------------------------------------------------------------------------------------------------------------------------------------------------------------------------------------------------------------------------------------------------------------------------|
| 1 | 3/14    | the authors should precise what
they mean by "flood risk". I guess
they
are referring to the "static" risk
of people being affected in their
home, as opposed to
dynamic flood risk, when people
are exposed during their
mobility.                                                                                                                                                                                                                                                                                                     | Yes, the Reviewer is correct. We inserted
the "i.e. to be affected in their home" to be
clear.                                                                                                                                                                                                                                                                                                                                                                                                                                                                                                                                                              |
| 2 | 3/18    | when the authors write
"motivations to protect
themselves", do they
mean to protect their property,
not their life?                                                                                                                                                                                                                                                                                                                                                                                                                                 | It's both. All the measures are meant to
protect the property, but at the same time
they can also save lives (e.g. having an
emergency plan or an adapted use of the
basement). That is why we kept a general
terminology.                                                                                                                                                                                                                                                                                                                                                                                                                         |
| 3 | 4/28    | I do not understand the last
sentence. Consequential – do you
mean
sequential?                                                                                                                                                                                                                                                                                                                                                                                                                                                                         | Yes, we mean sequential.                                                                                                                                                                                                                                                                                                                                                                                                                                                                                                                                                                                                                                          |
| 4 | 5/1     | "In these three studies" to
which studies does "these" refer
to?                                                                                                                                                                                                                                                                                                                                                                                                                                                                                          | They refer to the three studies that were
done in the context of natural hazards.
Therefore, we inserted "and more recently
applied by researchers (Bočkarjova et al.,
2009; Gebrehiwot and Veen, 2015; Martin et
al., 2007) in the natural hazard context"
prior to the sentence.                                                                                                                                                                                                                                                                                                                                                              |
| 5 | 5/30    | " of the public to undertake."                                                                                                                                                                                                                                                                                                                                                                                                                                                                                                                                  | Done.                                                                                                                                                                                                                                                                                                                                                                                                                                                                                                                                                                                                                                                             |
| 6 | 6/14-16 | could the authors provide more
information about the flood risk
maps. Are they built to map
areas at risk of floods of different
return periods? How is
the information about flood risk
communicated to people? This
could be an important
element for the interpretation of
the results of the study and I
believe it is important
to communicate this information
for people from countries where
flood risk information
could be different. In addition,
how do you explain that some
people had taken risk | Flood hazard maps are available for the main river system (the Adige) from the Distretto Idrografico Alpi Orientali (the former Water Authority). However, the resolution of these maps is not fine enough to capture flood hazards and risks generated by small river basins such as the Novare basin (2 km 2 ). These risks will be considered by novel maps (included in the Piano Comunale delle Acque), which are still in the making. Given the lack of a flood risk map at the required resolution, flood risk awareness is mainly based on historical evidences corresponding to floods which occurred in the last 80 years (the first flood). |

|   |         | reduction measures before the      | for which local data is available dates back                                     |
|---|---------|------------------------------------|----------------------------------------------------------------------------------|
|   |         | flood if the area was not declared | to May 17 1935.                                                                  |
|   |         | at risk?                           | This can be explained by the flood risk                                          |
|   |         |                                    | awareness mentioned above. Of course, the                                        |
|   |         |                                    | level of awareness and its potential to take                                     |
|   |         |                                    | risk reduction measures depends very much                                        |
|   |         |                                    | on personal experiences (either direct or                                        |
|   |         |                                    | indirect) and education.                                                         |
|   |         |                                    | We inserted this information in the                                              |
|   |         |                                    | manuscript: "This is due to the coarse                                           |
|   |         |                                    | resolution of these maps that are not fine                                       |
|   |         |                                    | enough to capture flood hazards and risks                                        |
|   |         |                                    | generated by small river basins such as the                                      |
|   |         |                                    | Novare basin (2 km 2 ). These risks will be                           |
|   |         |                                    | considered by novel maps, which are still in                                     |
|   |         |                                    | the making. Thus, flood risk communication                                       |
|   |         |                                    | is poor and flood risk awareness is mainly                                       |
|   |         |                                    | based on historical evidences corresponding                                      |
|   |         |                                    | to floods which occurred in the last 80 years.                                   |
|   |         |                                    | Nevertheless, some people have taken risk                                        |
|   |         |                                    | reduction measures in the past as the level                                      |
|   |         |                                    | of flood risk awareness and the readiness to                                     |
|   |         |                                    | take these measures depends, for instance,                                       |
|   |         |                                    | on personal experiences (either direct or                                        |
| 7 | c/2c 27 |                                    | Indirect) and education.                                                         |
| / | 6/26-27 | is there any warning to people     | A nood warning system is active in the                                           |
|   |         | when heavy precipitation is        | bazarda for specific targets in the region                                       |
|   |         | in a region? How were the losses   | whereas it provides generic warning for                                          |
|   |         | compensated if few people          | wide areas based on rainfall thresholds. For                                     |
|   |         | subscribe to a insurance           | a frazioni like Arhizzano only rainfall                                   |
|   |         | policy?                            | thresholds are available. Owing to the                                           |
|   |         | policy                             | limited extent (in space and in time) of the                                     |
|   |         |                                    | 2018 09 01 flash flood generating storm                                          |
|   |         |                                    | warnings were not issued. At the time of the                                     |
|   |         |                                    | survey, losses were not compensated vet                                          |
|   |         |                                    | (see comment 17). However, major losses                                          |
|   |         |                                    | can be compensated directly by the Veneto                                        |
|   |         |                                    | Region given formal documentation of flood                                       |
|   |         |                                    | damages.                                                                         |
|   |         |                                    | We inserted the following information:                                           |
|   |         |                                    | "Owing to the limited extent (in space and                                       |
|   |         |                                    | time) of the event, no (generic) flood                                           |
|   |         |                                    | warnings were issued, even though a                                              |
| 1 |         |                                    | , j                                                                              |
|   |         |                                    | threshold based warning system is in place.                                      |
|   |         |                                    | threshold based warning system is in place.
Thus, no preventive measures were |

| 8  | 6       | in the presentation of the case
study, could you also indicate the
types of houses
that were affected by the flood:
are they located in old or new
parts of the town? Do
the house have several floors (it
seems to be the case as
basement is mentioned p. 9
line 11)?                                                 | Both new and old parts of the town were
impacted by the flood. Palazzina
d'Arbizzano, which is the area with the
largest damages, is located in the oldest part
of the town.
Most of the damaged houses have several
(up to five) floors. We added the following
text:
"Both new and old parts of the town were
impacted by the flood. Palazzina
d'Arbizzano, which is the area with the
largest damages, is located in the oldest part
of the town. Most of the damaged houses
have several (up to five) floors."                                                                                                                                                                                                                 |
|----|---------|----------------------------------------------------------------------------------------------------------------------------------------------------------------------------------------------------------------------------------------------------------------------------------------------------------------------------------------------------|---------------------------------------------------------------------------------------------------------------------------------------------------------------------------------------------------------------------------------------------------------------------------------------------------------------------------------------------------------------------------------------------------------------------------------------------------------------------------------------------------------------------------------------------------------------------------------------------------------------------------------------------------------------------------------------------------------------------------------------------------------------------------|
| 9  | 7/16-17 | did some people mention other
protection measures than the
ones that are included in Table 1?                                                                                                                                                                                                                                                | No, people did not mention other measures.
There wasn't an open question to ask about
other protection measures. However we
prepared the list based on interviews and
discussion with local authorities and
residents.                                                                                                                                                                                                                                                                                                                                                                                                                                                                                                                                     |
| 10 | 7/24-28 | it could be easier for the reader
to present the percentages in
terms of increasing of decreasing
level of education. Furthermore,
the authors use
"compulsory school" and "middle
school": it is the same thing.                                                                                                                | We realized that the information in the
brackets was confusing and not relevant or
necessary for the understanding of the
information. Therefore, we deleted the text
in brackets.                                                                                                                                                                                                                                                                                                                                                                                                                                                                                                                                                                            |
| 11 | 8/1-14  | If you are able to affect the
different interviewees in the
three categories, it means that
they are aware that there is a risk
and are ready to protect
themselves. Are you sure of that?
Apparently not as you mention p.
9 line 14 that the majority of
people are not ready to
implement any protection
measure. | Yes, we believe, that people are at least to
some degree aware of the flood risk. The
sentence at p.9 line 14 is incorrect, or at
least the sentence structure is wrong. What
we initially meant to say is that the majority
of respondents only adopted a few of the
ten measures (or plan to do so in the near
future), while they would not implement
any of the other measures. Thus, it was also
possible to classify most of the people (124
of 146) into one of the three groups as they
adopted at least one or two measures (see
next paragraph). Therefore, we changed the
sentence into "Overall, it is important to
highlight that the majority of people
indicated to have implemented just a few
measures." |
| 12 | 8/20    | when you ask people if they
consider their house to be at risk
of flooding,                                                                                                                                                                                                                                                                  | We are referring to the perception of the
flood risk. Interviewees made that clear
when asking the questions (in Italian). We                                                                                                                                                                                                                                                                                                                                                                                                                                                                                                                                                                                                                                       |

|    |          | are you referring to their
perception of the risk or to
objective information like maps
of
flood prone areas?                                                                                                                                                                                          | exchanged "considered" with "perceived" to be clearer.                                                                                                                                                                                                                                                                                                                                                                                                                                                                                                                                                                                                                                                                                                                                                                                                                                                                                                              |
|----|----------|--------------------------------------------------------------------------------------------------------------------------------------------------------------------------------------------------------------------------------------------------------------------------------------------------------------------|---------------------------------------------------------------------------------------------------------------------------------------------------------------------------------------------------------------------------------------------------------------------------------------------------------------------------------------------------------------------------------------------------------------------------------------------------------------------------------------------------------------------------------------------------------------------------------------------------------------------------------------------------------------------------------------------------------------------------------------------------------------------------------------------------------------------------------------------------------------------------------------------------------------------------------------------------------------------|
| 13 | 8/31     | You mention the use of
regression analysis: does it mean
that all
the variables were transformed
to quantitative values? How did
you perform that for
"benefits"?                                                                                                                                | As we indicated a couple of sentences
above, we measured benefits with two
statements that referred to an extrinsic and
an intrinsic reward. It was measured on a 5-
point Likert scale from completely disagree
to completely agree.                                                                                                                                                                                                                                                                                                                                                                                                                                                                                                                                                                                                                                                                                                                |
| 14 | 9/14     | you mention that the majority of
people has not or did not plan to
perform protective measures.
How did you classify these people
in one of the three
categories?                                                                                                                                   | See comment 11.                                                                                                                                                                                                                                                                                                                                                                                                                                                                                                                                                                                                                                                                                                                                                                                                                                                                                                                                                     |
| 15 | 11/2     | Have you any idea of the results
of the survey if it has been
conducted before the flood?                                                                                                                                                                                                                    | No, not really. However, one could
speculate that the influence of PMT
variables in each group may not have
changed as experience was not found to be
significant.                                                                                                                                                                                                                                                                                                                                                                                                                                                                                                                                                                                                                                                                                                                                                                                      |
| 16 | 11/22    | "more important influence than threat"                                                                                                                                                                                                                                                                             | Done.                                                                                                                                                                                                                                                                                                                                                                                                                                                                                                                                                                                                                                                                                                                                                                                                                                                                                                                                                               |
| 17 | 11/26-27 | when you performed the survey,
had people already recovered
from the flood? Did some people
were affected on the long term?
Did they receive any
compensation for the damages?
Which type of communication
would be required
so that people remain aware of
the risk of flash flooding? | We did not explicitly ask these questions,
but from the interviews, it seemed that
almost all people recovered completely
from the flood. The majority had economic
damages, but no one had to permanently
leave their home. For instance, some people
had 3 meters of water in their basement,
but we do not know whether the basement
was again completely functional at the time
of the survey or not (e.g., due to mould).
One interviewee was getting flood
protection barriers installed by workers.
Regarding the psychological impact, that's
hard to say. It was a severe event, but
people did not die or were severely injured.
No, people did not receive any
compensation for damages at the time of
the survey.
We added some sentences: "Furthermore,
the flood may not have been severe enough,
as the damage was mostly economic, but no
respondents had to be relocated. As people |

|    |              |                                                                                                                                                                                                                                                                                                     | were not compensated for the damages,
they may have spent their money for
recovery instead of investing in future flood
preparedness. Also, (fortunately) no people
died or got severely injured, which may
further limit the psychological impact, and
thus the likelihood to engage in risk
reduction behaviours."
We describe the types of communication in                                                                                                                                                                                                                                                                                                                                                                                                                                                                                                                                                                                                                                                                                                                                                                                                                                                                                                                                                                                                                                                                           |
|----|--------------|-----------------------------------------------------------------------------------------------------------------------------------------------------------------------------------------------------------------------------------------------------------------------------------------------------|------------------------------------------------------------------------------------------------------------------------------------------------------------------------------------------------------------------------------------------------------------------------------------------------------------------------------------------------------------------------------------------------------------------------------------------------------------------------------------------------------------------------------------------------------------------------------------------------------------------------------------------------------------------------------------------------------------------------------------------------------------------------------------------------------------------------------------------------------------------------------------------------------------------------------------------------------------------------------------------------------------------------------------------------------------------------------------------------------------------------------------------------------------------------------------------------------------------------------------------------------------------------------------------------------------------------------------------------------------------------------------------------------------------------------------------------------------------|
|    |              |                                                                                                                                                                                                                                                                                                     | more detail in the second last paragraph.                                                                                                                                                                                                                                                                                                                                                                                                                                                                                                                                                                                                                                                                                                                                                                                                                                                                                                                                                                                                                                                                                                                                                                                                                                                                                                                                                                                                                        |
| 18 | 12/17-
18 | the last sentence is not clear.                                                                                                                                                                                                                                                                     | We understand that the explanation is not
clear, and as it is only our speculation, we
decided to delete the sentence.                                                                                                                                                                                                                                                                                                                                                                                                                                                                                                                                                                                                                                                                                                                                                                                                                                                                                                                                                                                                                                                                                                                                                                                                                                                                                                                                     |
| 19 |              | Discussion: could you add some
elements in the discussion
explaining how gen-
eral/particular are the results of
your case study? You should also
discuss the relevance of your
methodology as you mentioned
that you applied the PTT-TTM
models
differently than others | Based on the Reviewer's suggestion we
added a paragraph on the generalization at
the end of the discussion: "Finally, the
research was designed to analyse flood risk
awareness in Italy, and could therefore
provide insights that are relevant for
promoting the adoption of risk reduction
behaviours in small municipalities in the
Italian Alps. However, several case and
hazard-specific characteristics may hinder
result generalization. For instance, hazards
can be slower-onset, spatially diffuse (such
as droughts) or rapid-onset, spatially
localised (such as flash floods). This clearly
has relevant implications for preparedness,
including e.g. time availability to adopt risk
reduction behaviors. Furthermore,
preparedness is not only impacted by the
hazard characteristics, but also by social,
legal, cultural, institutional and political
aspects that can strongly differ from one
region/case to another. Thus, further
research would be needed to explore these
aspects, e.g. by applying the same PMT-
TTM model for different hazards and in
different countries."
In the first paragraph of the discussion, we
compare the general findings with the
other studies that applied PMT-TTM. We
added a couple of sentences. "Our findings
add to other studies in the natural hazard
field (Bočkarjova et al., 2009; Gebrehiwot |

|    |       |                                                                                                                                                                                               | 1                                                                                                                                                                                                                                                                                                                                                                                                                                                                                                                                                                                                                                                                                                                                                                                                                                                                                                                                                                                                                                                                                                                                                                                               |
|----|-------|-----------------------------------------------------------------------------------------------------------------------------------------------------------------------------------------------|-------------------------------------------------------------------------------------------------------------------------------------------------------------------------------------------------------------------------------------------------------------------------------------------------------------------------------------------------------------------------------------------------------------------------------------------------------------------------------------------------------------------------------------------------------------------------------------------------------------------------------------------------------------------------------------------------------------------------------------------------------------------------------------------------------------------------------------------------------------------------------------------------------------------------------------------------------------------------------------------------------------------------------------------------------------------------------------------------------------------------------------------------------------------------------------------------|
| 20 | 12-13 | what is presently in the conclusions, should appear in the                                                                                                                                    | and Veen, 2015; Martin et al., 2007), in
which people were grouped based on total
amount of their risk reduction behaviours.
However, we decided to classify people
based on the quality of these behaviours
and distinguished between structural,
avoidance, and emergency measures.
Interviews with local authorities confirmed
our belief that people who have an
emergency plan for their household may be
motivated by different factors than people
who keep expensive appliances above the
expected flood levels. The motivation is
again different for people who installed
anti-backflow valves in their houses.
Therefore, we can only compare the overall
results of the few studies that applied the
PMT-TTM models in the natural hazard
context, and not the specific findings for
each group." Furthermore, we discuss the
relevance of the model by emphasizing
specific implications for risk
communication strategies for different
people depending on their readiness (see
second last paragraph of discussion).
As suggested by the Reviewer, we moved
the part with the implications to the |
|    |       |                                                                                                                                                                                               | people depending on their readiness (see second last paragraph of discussion).                                                                                                                                                                                                                                                                                                                                                                                                                                                                                                                                                                                                                                                                                                                                                                                                                                                                                                                                                                                                                                                                                                                  |
| 20 | 12-13 | what is presently in the
conclusions, should appear in the
discussion and
the conclusions should include
the main findings of the study,
not possible outcome of
the study. | As suggested by the Reviewer, we moved
the part with the implications to the
discussion (see second last paragraph of
discussion). We summarized the key
findings in the conclusion. We end with the
limits of our approach.                                                                                                                                                                                                                                                                                                                                                                                                                                                                                                                                                                                                                                                                                                                                                                                                                                                                                                                                                     |